# Photo-Functionalized Magnetic Nanoparticles as a Nanocarrier of Photodynamic Anticancer Agent for Biomedical Theragnostics

**DOI:** 10.3390/cancers12030571

**Published:** 2020-03-01

**Authors:** Ki Chang Nam, Yong Soo Han, Jong-Min Lee, Si Chan Kim, Guangsup Cho, Bong Joo Park

**Affiliations:** 1Department of Medical Engineering, Dongguk University College of Medicine, Gyeonggi-do 10326, Korea; kichang.nam@gmail.com; 2Department of Radiological Science, Hallym Polytechnic University, Chuncheon 24210, Korea; yonggali80@naver.com; 3Lugen Bio R&D Center, Lugen Sci. Co. Ltd. 397, Seokcheon-ro, Bucheon-si 14449, Korea; jmlee@lugensci.com; 4Department of Electrical & Biological Physics, Kwangwoon University, 20 Kwangwoongil, Nowon-gu, Seoul 01897, Korea; kaja7776@naver.com (S.C.K.); gscho@kw.ac.kr (G.C.); 5Institute of Biomaterials, Kwangwoon University, 20 Kwangwoongil, Nowon-gu, Seoul 01897, Korea

**Keywords:** photodynamic therapy, multifunctional magnetic nanoparticles, magnetic resonance imaging, fluorescence cellular imaging, anticancer activity, apoptotic cell death, biomedical theragnostic

## Abstract

Various theragnostic agents have been devised and developed as cancer treatments; however, existing agents are often limited by their specific functions and complexities. Here, we report multifunctional magnetite (Fe_3_O_4_) nanoparticles functionalized with chlorin e6 (Ce6) and folic acid (FA) using a simple fabrication process to be used as theragnostic agents in photodynamic therapy (PDT). The effectiveness of cellular uptake of Fe3O4-Ce6-FA nanoparticles (FCF NPs) and its visualization as well as the photodynamic anticancer activities were evaluated. The mechanism of cancer cell death by the FCF NPs was also verified with qualitative and quantitative methods. Results indicate that FCF NPs have good penetration efficacy, resulting in excellent in vitro fluorescence and magnetic resonance imaging in cancer cells. FCF NPs exhibited promising anticancer activity in an irradiation time- and FCF NPs-dose-dependent manner in various cancer cell lines, leading to apoptotic cell death via morphological changes in cell membrane, nuclear, and DNA damage, and via overexpression of apoptosis-related genes, such as ZFP36L1, CYR61, GADD45G, caspases-2, -3, -9, 10, and -14. This study suggests that FCF NPs may be safely used in cancer therapy via PDT and could be a versatile therapeutic tool and biocompatible theragnostic agent, which may be used in diagnostic imaging.

## 1. Introduction

Theragnostics, which integrate diagnostic imaging ability and therapeutic intervention in a single delivery system, have gathered increasing attention in both biomedical research and clinical fields [1,2]. Moreover, the utilization of theragnostic agents with various functions enables simultaneous disease diagnosis and targets detection, real-time monitoring of drug distribution and therapeutic progress, as well as the efficacy of therapy [1,2,3]. Therefore, theragnostics are expectedly a very effective treatment, enabling biomedical technology to achieve modern personalized and precision medicine [2,3,4].

Most of the theragnostic-based approaches require non-invasive imaging abilities, such as magnetic resonance (MR) imaging, computed tomography, positron emission tomography, and ultrasonic imaging to achieve real-time monitoring of biodistribution, release kinetics, treatment progress, and efficacy of their agents [5].

Among various imaging technologies for theragnostic, MR imaging is currently one of the most broadly used and powerful and is used not only in clinical diagnosis but also in biomedical fields as a tool for non-invasive clinical diagnosis [6]. MR imaging has numerous advantages such as non-invasiveness, a high degree of soft-tissue contrast, deep tissue visualization, sub-millimeter spatial resolution, and non-ionizing radiation [7,8].

Recently, advanced nanotechnology may be applied to enhance the efficacy of theragnostic [9,10,11]. Especially, magnetic nanoparticles (MNPs), polymeric nanoparticles (NPs), lipid-based NPs, dendrimers, cage proteins, and inorganic NPs are nanomaterials that are most commonly used in theragnostic [9,10,11,12,13,14,15,16,17,18,19,20,21,22,23]. Among these, MNPs have been widely applied in biomedical diagnostics such as magnetic resonance imaging (MRI), as well as in therapeutics [24,25], cancer therapy [23,24], hyperthermia [25,26], drug and gene delivery [27,28], tissue engineering [29,30,31], biosensors [32,33] and other aspects [34,35,36,37]. The wide applicability of MNPs is attributed to the many advantages associated with their use, such as a large surface area to volume ratio for loading drugs or genes [25,26], superparamagnetic properties [24,25] and biocompatibility. Furthermore, MNPs have attracted much attention as the most extensively applied MNPs, owing to their excellent chemical stability, mechanical hardness, high magnetic properties, biocompatibility and lower cytotoxic profile, based on the multifunctionality of iron oxide [11,12,13,14,15,24,25,26,27,28]. Therefore, superparamagnetic iron oxide nanoparticles (SPIONs) are widely applied for MRI as a T_2_-weight contrast agent as they are not subject to strong magnetic interactions during dispersion, and are readily stabilized under physiological conditions [38].

In recent times, scientists have focused on designing and fabricating MNPs that are combined or conjugated with various functional materials associated with fluorescence, plasmonics, magnetism, and bio-targeting ability, among others to achieve the functionality as theragnostic agents [39,40,41,42].

We have additionally previously synthesized two kinds of multifunctional MNPs: cobalt ferrite nanoparticles functionalized with hematoporphyrin (HP) for introducing photo-functionality and folic acid (FA) for targeting cancer cells [39], and iron (Fe_3_O_4_) NPs functionalized by conjugation with chlorin e6 (Ce6) for photo-functionality and FA as a target molecule for cancer cells [43]. Previous studies aimed at fabricating novel MNPs with anticancer activities via photodynamic therapy (PDT). We also recently reported the material properties and PDT activity of Fe3O4-Ce6-FA nanoparticles (FCF NPs) in two cancer cell lines: breast cancer (MCF-7 cells) and prostate cancer (PC-3 cells) [43]. 

Here, we focused on in vitro FCF NP cellular uptake via atomic absorption spectroscopy (AAS) analysis, fluorescence imaging, and MR imaging in 4 cancer cells, cervical (HeLa), breast (MCF-7), prostate (PC-3), and ovarian (SKOV-3) cancer cells, to confirm its potential as an image-guided theragnostic agent for PDT. We then investigated the photodynamic anticancer activities of FCF NPs, against four cancer cell lines to verify the potential of nanocarriers for PDT. Finally, we examined the mechanism underlying cancer cell death using Ampli-Seq sequencing to analyze the differentially-expressed mRNA related to apoptotic cell death, morphological changes, and Caspase-3/7 activities with qualitative and quantitative methods to determine the potential utility in clinical applications.

## 2. Results and Discussion

### 2.1. Cellular Uptake of FCF NPs by MCF-7 Cells

Twenty-nanometer, multifunctional magnetic nanoparticles (FCF NPs), with a long-wavelength absorption band and high singlet oxygen quantum yield, were prefabricated using a simple surface modification, as shown in Figure 1, and as previously described [43,44,45].

Cellular uptake of FCF NPs by cancer cells is considered important for the anticancer effects of PDT. Therefore, AAS analysis was utilized to quantify the FCF NPs in each cancer cell by detecting iron content (Fe^2+^ or Fe^3+^) following treatment with 20 µg/mL FCF NPs for 2 h. The amount of FCF NPs in each cancer cell was about 34.6 ± 2.84 pg/cell for HeLa, 32.0 ± 2.98 pg/cell for MCF-7, 39.0 ± 3.8 pg/cell for PC-3, and 46.7 ± 5.2 pg/cell for SK-OV3, respectively. In addition, we also measured the fold change (FC) NP concentration and unconjugated FA to evaluate the FA targeting efficiency of cancer cells. The FC NP amount in each cancer cell reached about 65–74% (36.2 ± 3.44 pg/cell for HeLa, 20.8 ± 4.88 pg/cell for MCF-7, 28.9 ± 4.12 pg/cell for PC-3, and 34.6 ± 3.22 pg/cell for SK-OV3, respectively). These results indicate that FCF NPs entered into the cytoplasm of cancer cells more readily than FC NPs. Further, these results show excellent anticancer activity with high FCF NP concentration via PDT in cancer cell lines.

Based on the AAS analysis results, four green fluorescent protein (GFP)-transfected cancer cell lines [46] were used to visualize FCF NPs in the cytosol of each cancer cell after incubation for 2 h with 20 μg/mL FCF NPs. High-intensity orange fluorescence was detected in all cancer cells after 2-h incubation. The distribution of orange fluorescence in the cytoplasm of each cell was relatively uniform (Appendix A). The fluorescence intensity in each cell type was also confirmed via quantitative fluorescence analysis (data not shown). The results of fluorescence intensity analyses were similar to fluorescence cellular imaging results. However, orange fluorescence was not detected in untreated control cells. This result indicates that fluorescence-based cellular images were closely correlated with the AAS data. Additionally, the data showed good FCF NP cellular uptake. Furthermore, this result indicates that 2 h incubation is sufficient for cellular FCF NP uptake for PDT.

TEM analysis was conducted using MCF-7 cells following incubation with 20 µg/mL of FCF NPs for 1, 2, and 4 h, to determine the internalization and optimal period of incubation for cellular uptake.

The number of FCF NPs in MCF-7 cells gradually increased with the duration of incubation; however, most FCF NPs were located in the cytosol of MCF-7 cells, regardless of incubation time (Figure 2a).

Cellular TEM images clearly demonstrated that the FCF NPs were absorbed well by MCF-7 cells and that most FCF NPs were present in the cytosol. These results indicate that the absorption mechanism of FCF NPs involved FA receptor-mediated endocytosis: MCF-7 cells express the FA receptor protein on their cell membrane surfaces [47], and the FA receptor binds the FCF NPs, resulting in their intracellular uptake.

Further, the amount of FCF NPs in MCF-7 cells increased with incubation in a time-dependent manner. This result was closely consistent with that of our previous reports [39,43,45]. In these previous studies, it was shown that an increase in the time that MNPs were incubated with cancer cells resulted in a significant increase in the amount of MNPs in cancer cells, leading to significantly improved anticancer activity at 2 and 4 h of incubation by PDT compared with that achieved after 1 h of incubation. However, there was no big difference in PDT activity at 2 h and 4 h of incubation with the FCF NPs (Figure 2b), as previously reported [45]. Therefore, 2 h was selected as the optimal duration of incubation of PDT with FCF NPs.

### 2.2. In Vitro MR Imaging of FCF NPs in Various Cancer Cells

Magnetic resonance (MR) imaging was conducted to evaluate the potential of FCF NPs as T_2_-type contrast agents. This evaluation was considered important as MR imaging is a powerful imaging tool that is widely used for non-invasive diagnoses owing to its spatial resolution and depth of tissue penetration [7]. Iron oxide nanoparticles have long been used as T_2_ contrast agents [48,49]. To analyze in vitro MR imaging via FCF NPs, MRI contrasts of four cancer cell types were detected following incubation with concentrations of 0 to 220 µg/mL of FCF NPs for 2 h, using a 3.0 tesla MRI instrument. T_2_ signal intensities of phantom images in each cancer cell exhibited progressively dose-dependent contrast, as shown in the inner box in Figure 3. Moreover, the relaxation time (T_2_) was also obtained to analyze the ability of T_2_-MR imaging by FCF NPs. Appendix A shows the T_2_ relaxation rate (1/ T_2_); the maximum T_2_ relaxation rate was observed at a concentration of 20 µg/mL. The reciprocal values for the T_2_ relaxation rate at the concentration of 20 µg/mL (5.77, 4.18, 4.40, and 5.24 s-1) are 173, 239, 272, and 190 ms for HeLa, MCF-7, PC-3, and SKOV-3 cells, respectively. Figure 3 also shows the plot of the T2 relaxation rate, which depicts a linear increase. These results indicate that FCF NPs may be useful as a theragnostic agent for MR imaging, as they exhibit strong magnetization. Further, the T_2_ signal intensities of phantom images for each cancer cell changed gradually depending on the concentrations of the FCF NPs, resulting in a close correlation between MR imaging ability and anticancer efficacy. Additional studies involving in vivo experiments are warranted.

### 2.3. In Vitro Cytotoxicity of FCF NPs

For the purpose of confirming cytotoxicity of FCF NPs, cytotoxicity testing was performed in each of the five cell types, namely L-929, HeLa, MCF-7, PC-3, and SKOV-3, using a method recommended by the International Organization for Standardization (ISO 10993-5), as previously described [35,39,40,41,42]. A significant reduction in viability due to treatment with FCF NPs for 24 h was not observed (Figure 4). Furthermore, cellular images using GFP-transfected cells showed no changes in their signal intensity, indicating that most cells were viable (Appendix A). The viability of each cell type was shown to be over 92%. These results indicate that FCF NPs exhibit no cytotoxicity, as previously reported [35,39,40,41,42], and therefore show potential for clinical applications.

### 2.4. In Vitro Photodynamic Anticancer Activity of FCF NPs

To compare the PDT effects of Ce6 and NPs, the photodynamic properties of Ce6 in cancer cells were also detected as shown in Appendix A. The amount of Ce6 on the surface of the FCF NPs was determined using UV-Vis spectroscopy and calculated as about 0.375 µg per 25 µg of Fe_3_O_4_ NPs. This data using only Ce6 in cancer cells reveal that no PDT effects occurred with the amount of Ce6 (0.375 µg) used; however, the PDT data for the FC and FCF NPs (Appendix A) exhibited more strong photodynamic anticancer activities than that of Ce6. Especially, the FCF NPs showed excellent anticancer activity (Appendix A). These results indicate that the FCF NPs taken up into cytosol can produce high singlet oxygen by PDT, leading to cancer cell death.

The most important factors that determine the efficacy of PDT are the dose of the photosensitizer and the dose of light. The photosensitizer dose is obviously correlated with FCF NP concentration. Therefore, the effects of the dose of light were examined in this study.

To determine whether anticancer activity of PDT in the cancer cells (HeLa, MCF-7, PC-3, and SKOV-3) was dependent on the duration of exposure to light-emitting diode (LED) light, cells were incubated with various concentrations of FCF NPs for 2 h and exposed to a 40 mW LED light source for 10, 20, and 30 min. In order to confirm photodynamic efficacy, the viability of each cell type was analyzed using the CCK-8 method, as shown (Figure 5). Regardless of cell type, anticancer activities of FCF NPs were significantly increased depending on LED irradiation time: even 10-min irradiation with LED dramatically decreased the viability of all cells, depending on FCF NP concentration.

At a concentration of 12.5 μg/mL of FCF NPs, prostate cancer (PC-3) cells that were LED-irradiated for 20 and 30 min showed a marginally higher photodynamic killing efficacy than the other cancer cells, HeLa, MCF-7 and SKOV-3. However, there were no major differences in the viability of all cells following 10 min LED irradiation. These results demonstrated that a 10-min LED exposure was sufficient to induce the effects of photodynamic anticancer activity on any cancer cell. Furthermore, PDT efficacy was closely correlated to both exposure time to LED light and the dose of FCF NPs.

### 2.5. Analysis of FCF NP-Induced Apoptotic Cell Death in Cancer Cells

We next investigated the mechanism of cancer cell death by analyzing (1) mRNA expression using Ampli-Seq sequencing, (2) Caspase-3/7 enzyme activity, (3) phosphatidylserine translocation in cell membranes, (4) nuclear fragmentation, and (5) DNA damage in HeLa cells.

First, we verified the differentially-expressed genes related to apoptotic cell death in Hela cells. Among the 827 cell death-related genes, 590 genes were identified. Differential analysis revealed that the differentially expressed genes were categorized into three terms: cell death, apoptotic process, and apoptotic mitochondrial changes. Seventeen total genes were related to cell death, fourteen genes to the apoptotic process and two genes for apoptotic mitochondrial changes were identified (cutoffs: FC > 1.5, Figure 6a).

Among the differentially expressed genes related to cell death, most were apoptotic process-related genes (Figure 6a,c,d). The significantly upregulated genes were zinc finger protein 36, C3H type-like 1 (ZFP36L1), cysteine-rich angiogenic inducer 61 (CYR61), and growth arrest and DNA-damage-inducible protein 45 gamma (GADD45G). Among these, ZFP36L1 was the most dramatically over-expressed. The highly-expressed ZFP36L1, CYR61, and GADD45G genes play an important role in the induction of apoptotic cell death.

The ZFP36L1 gene regulates various cellular processes, including cell apoptosis, by binding to adenine/uridine rich elements in the 3ʹ untranslated regions of target mRNAs, which promotes their degradation [50,51,52]. The CYR61 gene induces apoptosis and cell senescence [53,54,55], which are well-established tumor suppression mechanisms [56]. Furthermore, by inducing apoptosis, tumorigenesis can be suppressed, thereby preventing the proliferation of damaged cells [57]. Moreover, apoptosis is increased in these cells in the presence of TNF-related apoptosis-inducing ligand [57]. Thus, the CYR61 gene can promote cancer cell death via apoptosis. The GADD45G gene is also a positive mediator of apoptosis and growth arrest in various human cancer cells, along with CR6, DDIT2, GRP17, OIG37, and GADD45gamma [58,59,60,61,62,63,64].

Based on the sequencing results of these genes, ZFP36L1, CYR61, and GADD45G overexpression (Figure 6) contribute to the induction of apoptosis in response to PDT after FCF NP treatment. Further, PDT using FCF NPs may affect transcriptional regulation of the cancer cell death pathway via the initiation of apoptosis in cancer cells.

Caspase genes are grouped into two subfamilies, either pro-apoptotic (caspase-2, -3, -6, -7, -8, -9, -10, and -14) or pro-inflammatory (caspase-1, -4, -5, -11, and -12) subfamilies. These genes encode proteins related to the maintenance of homeostasis via regulation of apoptotic cell death and inflammation [65,66,67]. The pro-apoptotic caspases are mediators of the apoptosis signal transduction, while pro-inflammatory caspases regulate the maturation of various cytokines during inflammation [67]. Pro-apoptotic caspase genes into two groups. One is initiator caspases (caspase-1, -2, -4, -5, -8, -9, -10, -11, and -12), and the other is effector caspases (caspase-3, -6, -7) [65,67]. Initiator and effector caspase expression in cancer cells can induce apoptosis.

Differentially-expressed apoptotic caspases genes were also identified via Ampli-Seq analysis (Figure 7a). Among the expressed caspase genes, caspase-2, -3, -9, -10, and -14 were significantly over-expressed. Importantly, two initiator caspases, caspase-2 and -9, and an executioner caspase, caspase-3, were more highly expressed than other caspase genes. Moreover, caspase-14 was also significantly increased after 1 h incubation post-PDT. These results indicate that caspase gene expression related to apoptotic cell death may induce apoptosis in cancer cells.

Because caspase-3/7 enzymes are hallmarks of apoptosis, we analyzed caspase-3/7 enzyme activity using a luminescence quantification assay. Caspase-3/7 activity at incubation periods of 1, 2, 4, and 8 h following PDT are shown (Figure 7b). Compared with that in the control, caspase-3/7 activity increased significantly with incubation time. These data also indicate that high levels of caspase-3/7 expression may result in apoptotic cell death in HeLa cells.

Finally, we analyzed morphological membrane changes to evaluate HeLa cell apoptosis after 10-min PDT for 10 min after 2-hr incubation with 25 μg/mL FCF NPs. The translocation of phosphatidylserine in the plasma membrane, a hallmark of early apoptosis in the HeLa cells, was confirmed via staining with the Annexin V-fluorescein isothiocyanate (FITC) apoptosis detection kit. FITC-labelled Annexin V can be used to specifically target and identify translocated phosphatidylserine in apoptotic cells. Fluorescence images of control and apoptotic HeLa cells following a 4 h post-irradiation incubation period are shown (Figure 7c). Strong green fluorescence signals were detected in cells treated with FCF NPs; however, no such signals were detected in untreated control cells. This finding indicates that LED irradiation following incubation with FCF NPs mediates phosphatidylserine translocation in the membrane of HeLa cells, leading to the early stages of apoptotic cell death.

Figure 3 activity in HeLa cells were also examined via fluorescence images obtained using Hoechst 33342 dye and a CellEvent Caspase-3/7 Green Detection reagent. Following incubation with FCF NPs for 2 h, the nuclei of LED-irradiated cells condensed gradually and developed irregular apoptotic granular nucleus bodies, whereas the controls did not display such changes (Figure 7d). Moreover, the cellular images showing green fluorescence revealed a strong expression of caspase-3/7 enzyme in cells treated with FCF NPs; in contrast, green fluorescence signals were not detected in control cells (Figure 7d).

Finally, DNA damage caused by LED irradiation was determined using an alkaline comet assay (Figure 7e,f). The images of green fluorescence comets in HeLa cells treated with FCF NPs indicated that the tails of comets in these cells were longer than in the control. Moreover, the movement of the tails, as well as percentage of DNA in the tails of cells treated with FCF NPs, were dramatically increased compared with those of control cells. Furthermore, comet assay data indicated that LED irradiation may cause a breakage of DNA strands in HeLa cells, leading to apoptotic cell death.

Apoptosis data (Figure 7) were closely consistent with PDT activity of FCF NPs (Figure 5 and Figure 6), as also described in our previous report [39].

Taken together, in vitro results showing apoptotic cell death indicate that our FCF NPs, which were fabricated using a simple method, exhibit excellent potential as a theragnostic agent with cancer targeting and monitoring inducing apoptosis in various cancer cells via PDT. However, in vivo tumor model studies to examine biological and physiological activity may be needed to verify its potential prior to use in clinical trials.

## 3. Materials and Methods

### 3.1. Preparation and Characterization of FCF NPs

Multifunctional FCF NPs (20 nm in size) were fabricated via a simple surface modification process (Figure 1a), based on a previously reported procedure with modification [39,43,44,45]. Briefly, 0.54 g of FeCl3·6H2O and 1.5 g of NaAc·3H2O were mixed with ethylene glycol (EG), and diethylene glycol (DEG) (1:19, υ:υ) mixture solution and the mixture was strongly stirred for 30 min. Then, the homogeneous mixture solution was heated and kept at 200 °C for 10 h with sealed in a stainless-steel autoclave. Precipitation in black was dried at 60 °C for 12 h after harvesting by magnetic decantation and washing with distilled water and absolute alcohol several times.

For introducing photoactive functionality into the Fe3O4 nanoparticles, a wet chemical process was used as previously described [39,43,44,45]. In brief, Fe3O4 nanoparticles (20 nm size) were mixed with Ce6 and ethanol solution (final concentration, 10−4 M). The Ce6 solution with Fe3O4 nanoparticles was vigorously stirred for 24 h at room temperature (R.T), and the Fe3O4 nanoparticles conjugated with Ce6 (Fe3O4-Ce6, FC) were washed several times with absolute alcohol. Applying the process described above, Ce6, which has three terminal carboxyl groups, a long-wavelength absorption, and can produce high levels of singlet oxygen, was easily conjugated with the Fe3O4 nanoparticles by esterification reactions. The amounts of Ce6 on the surface of the Fe3O4 nanoparticles were measured using UV-Vis absorption spectroscopy. The amounts of Ce6 molecules bound to the surface of Fe3O4 nanoparticles (6.25, 12.5, 25, 50, and 100 µg) were 0.09, 0.19, 0.38, 0.75, and 1.50 µg, respectively.

For targeting functionality of the 20 nm-sized Fe3O4-Ce6, the Fe3O4-Ce6 was dispersed in FA and dimethylsulphoxide (DMSO) solution and the solution was agitated for 5 h at R.T. Next, the Fe3O4-Ce6 and FA composites were harvested by magnetic decantation and washed using DMSO solution and phosphate-buffered saline several times. Finally, the washed nanoparticles were fully dried. The amounts of FA molecules on the Fe3O4 nanoparticles were also measured using UV-Vis absorption spectroscopy. The amounts of FA molecules on the surface of Fe3O4 nanoparticles (6.25, 12.5, 25, 50, and 100 µg) were 0.03, 0.06, 0.12, 0.24, and 0.47 µg, respectively.

The morphology of the FCF NPs was confirmed with a transmission electron microscope (TEM, JEM-2100F, JEOL, Tokyo, Japan) and field-emission scanning electron microscope (FE-SEM, SU-70, Hitachi, Tokyo, Japan), as shown in Figure 1b.

### 3.2. Cells and Cell Cultures

Five cell lines, mouse fibroblast (L-929), human cervix adenocarcinoma (Hela), human breast adenocarcinoma (MCF-7), human prostate cancer cell (PC-3), and ovary adenocarcinoma (SKOV-3), were purchased from the American Type Culture and Collection. L-929, Hela, MCF-7, and SKOV-3 cells were maintained in Dulbecco’s modified Eagle’s medium (DMEM) containing 10% fetal bovine serum (FBS) and 1% antibiotic antimycotic solution (Sigma-Aldrich Co., St. Louis, MO, USA). PC-3 cells were cultured in RPMI-1640 with 10% FBS and 1% antibiotic antimycotic solution. All cell lines were maintained at 37 °C in 5% CO_2_ and 95% air.

### 3.3. Analysis of Cellular Uptake of FCF NPs in Cancer Cells

To quantify intracellular FCF NPs, we measured cellular iron content in cancer cells. Pre-cultured HeLa, MCF-7, PC-3, and SKOV-3 cells were seeded in 24-well plates at a density of 1.5 × 10^5^ cells per well and incubated for 24 h at 37 °C (5% CO_2_ and 95% air). After incubation, the cells were refreshed with new media and incubated for another 2 h with 20 µg/mL FCF NPs in the dark. The cells were washed three times with Dulbecco’s phosphate-buffered saline (DPBS), treated with trypsin/EDTA, and collected to quantify the intracellular FCF NP uptake using an AAS.

To analyze the intracellular FCF NP, we used fluorescence imaging to detect FCF NP fluorescence in GFP-transfected cancer cells, as previously described [46].

All cancer cell lines used in this study were transfected with GFP. The cellular morphology of each line was visualized using a fluorescence microscope to confirm the possibility of FCF NP fluorescence imaging, as previously described [43]. Briefly, all GFP-transfected cancer cells were plated in 24-well plates [µ-plate 24-well ibiTreat (82406), ibidi GmbH, Germany] at of 0.5 × 10^5^ cells per well and incubated for 24 h at 37 °C (5% CO_2_ and 95% air). Next, the cells were treated with 20 µg/mL FCF and incubated for 2 h in the dark. Cells were washed three times with DPBS. Cellular morphology was observed using a live cell imager (LIONHEART FX; BioTek Instruments, Inc., Winooski, VT, USA) with a 40x objective lens and fluorescence optics (excitation/emission at 469/525 nm and 400/647 nm for GFP and Ce6, respectively).

Furthermore, FCF NP cellular uptake and intracellular localization were also evaluated in MCF-7 cells via TEM analysis, as previously described [39,43,44,45]. Briefly, pre-cultured MCF-7 cells (1.5 × 10^5^ cells per well) in a 24-well plate for 24 h at 37 °C in 5% CO_2_ and 95% air were treated with FCF NPs (20 µg/mL) and incubated for 1, 2, and 4 h in the dark, to determine optimal cellular uptake time. At each time point during incubation, each MCF-7 cell was fixed with 2.5% glutaraldehyde and 4% formaldehyde after washing three times with DPBS. Fixed cells were treated with 2% osmium tetroxide for 2 h at 4 °C, following washing twice with DPBS. Next, the cells were dehydrated with ethanol and embedded in Araldite. Finally, the cells were observed using a TEM (JEM-1011, JEOL).

### 3.4. In Vitro MR Imaging of FCF NPs in Various Cancer Cells

In order to perform MR imaging of FCF activity in cancer cells, HeLa, MCF-7, PC-3, and SKOV-3 pre-cultured cancer cells at concentrations of 1.5 × 10^5^ cells per well, each, were allowed to react with 0 to 220 µg/mL concentrations of FCF NPs for 24 h in a 24-well plate. Following a 2 h incubation period, each cell line was prepared for MRI as previously described [39,43,44,45]. Briefly, cells of each cell line were collected by centrifugation after washing three times with DPBS and detaching with trypsin/EDTA. Collected cells were resuspended in 2 mL tubes containing 1.5% agarose solution. Subsequently, fixed cells were observed using a 3.0 Tesla MRI instrument (SIEMENS MAGNETOM Skyra, A Tim + Dot system, USA). The T_2_-weighted images were acquired via conventional spin-echo acquisition (TR = 2210 ms, TE = 13.8, 27.6, 41.4, 55.2, and 69 ms, FOV = 220 × 220, slice thickness = 3 mm, matrix 384 × 384, acquisition number = 10).

### 3.5. In Vitro Cytotoxicity of FCF NPs

In order to evaluate its safety in vitro, cytotoxicity tests were performed on mouse fibroblast cells, L-929, and the human cancer cells, Hela, MCF-7, PC-3, and SKOV-3, as previously described [39,43,44,45]. Briefly, similar concentrations of pre-cultured cells were plated in 24-well plates (Coster Corp., MA, USA) under similar culture conditions for 24 h, as described above, to analyze cellular uptake. All cells were treated with 5 concentration levels of FCF NPs (0, 6.25, 12.5, 25, 50, and 100 µg/mL) and further incubated for 24 h at 37 °C (5% CO_2_ and 95% air) under dark conditions. Cells were incubated with 200 µL of culture media, including 20 µL of a Cell Counting Kit-8 (CCK-8) solution (Dojindo Laboratories, Kumamoto, Japan) for 15 min after washing three times with DPBS. Cell viability of each line was calculated by measuring the optical density of each well at 450 nm with a multi-mode microplate reader (Cytation 3; BioTek Instruments, Inc., Winooski, VT, USA). Cell viability was estimated as the percentage of live cells relative to control (untreated) cells. The morphology of cells treated with FCF NPs for 24 h was analyzed using a live cell imager (LIONHEART FX, Winooski, VT, USA).

### 3.6. In Vitro Photodynamic Anticancer Activity of FCF NPs

To evaluate the association between the PDT effect of FCF NPs and the duration of exposure to light, pre-cultured human cancer cells, Hela, MCF-7, PC-3, and SKOV-3, seeded at concentrations of 1.5 × 10^5^ cells per well in a 24-well plate for 24 h, were incubated with 0, 6.25, 12.5, 25, 50, and 100 µg/mL of FCF NPs for 2 h in the dark. After the cells were washed three times with DPBS, the culture medium was changed. Next, the cells were irradiated with a red LED for 10, 20, and 30 min as previously described [43] The LED had a maximum wavelength of 660 nm, and the irradiation power was 40 mW/cm^2^. LED-irradiated cells were further incubated for 24 h, and cell viabilities measured using the same CCK-8 kit as described above.

For cellular imaging, cancer cells, GFP-transfected human cancer cells, Hela, MCF-7, PC-3, and SKOV-3, following LED irradiation, were also cultured with similar concentrations of FCF NPs for 2 h at 37 °C (5% CO_2_ and 95% air) in the dark, irradiated with LED for 10 min, and further incubated for 24 h. Subsequently, fluorescence images of live cancer cells were acquired with an automated live-cell imager (LIONHEART FX) with 10x objective lens and fluorescence optics (excitation at 377 nm for GFP and at 469 nm for FCF, and emission at 447 nm for GFP and at 525 nm for FCF).

### 3.7. Detection of Apoptotic Cell Death by PDT

To confirm the mechanism of cell death from PDT via FCF NPs, we performed 5 assays. We investigated (1) differential mRNA expression related to apoptotic cell death, (2) phosphatidylserine translocation in the plasma membrane, (3) nuclear fragmentation, (4) DNA damage, and (5) caspase-3/7 activity in HeLa cells, as previously described [39,43]

First, we used Ampli-Seq sequencing to evaluate the gene expression within cancer cells after PDT. Briefly, total RNA was isolated from HeLa cells 0 and 60 min after 10 min incubation with PDT using Trizol solution (Invitrogen). The total extracted RNA quality and quantity were analyzed with a Bioanalyzer (Agilent 2100, Agilent, Santa Clara, CA, USA) using an RNA Nano Chip (RNA 6000, Agilent Technologies, Amstelveen, Netherlands) and a NanoDrop spectrophotometer (ND-2000, Thermo Inc., DE, USA). Ampli-Seq libraries were constructed and sequenced on an Ion Torrent S5xl platform (Thermo Fisher Scientific, Vista, CA, USA). To analyze human genes, an Ion Ampli-Seq Transcriptome Human Gene Expression Kit including over 20,000 human Ref-Seq genes, was used to prepare cDNA libraries from 30 ng extracted total RNA from each test sample. Multiple libraries were also clonally amplified using the Ion Chef System (Thermo Fisher Scientific) and sequenced on an Ion Torrent S5xl instrument. The differentially expressed genes were defined as genes with at least 1.5-fold change and an adjusted *p*-value < 0.1 (Benjamini-Hochberg correction).

Second, to detect phosphatidylserine translocation in the plasma membrane, pre-cultured HeLa cells (1.5 × 10^5^ cells per well) were incubated with 25 μg/mL FCF NPs for 2 h and then incubated for a further 4 h at 37 °C (5% CO_2_ and 95% air) following irradiation with LED for 10 min. Then, cells in each well were treated with an EzWay Annexin V-FITC apoptosis detection reagent (K29100, Komabiotech Inc., Seoul, Korea) for staining of translocated plasma membranes. The translocation of phosphatidylserine across the membranes of HeLa cells following PDT was observed with an automated live-cell imager (LIONHEART FX) as described above.

Third, in order to analyze nuclear fragmentation, HeLa cells incubated for 4 h following LED were irradiated for 10 min post-incubation and treated with 20 μg/mL of the FCF NPs for 2 h. Then, cells were fixed and stained with Hoechst 33342 (H3570, Invitrogen, Molecular Probes, Eugene, OR, USA), which stains cell nuclei. The stained nuclei of HeLa cells were observed and fluorescence images acquired using an automated live-cell imager (LIONHEART FX) with a 60x objective lens. The wavelengths of fluorescence optics were 399 nm for excitation and 447 nm for emission.

Fourth, a fluorescence imaging assay was conducted to detect and visualize Caspase-3/7 activity as follows; briefly, HeLa cells irradiated with LED for 10 min, following incubation with 20 μg/mL of FCF NPs for 2 h; then, cells were further incubated for 4 h and treated with CellEvent Caspase-3/7 Green Detection reagent (Thermo Fisher Scientific, Carlsbad, CA, USA) for 30 min, to stain activated Caspase-3/7 enzyme. After fixing with 70% EtOH and drying at room temperature (RT) in the dark, the stained cells were observed using an automated live-cell imager (LIONHEART FX) with a 60x objective lens. As described above, fluorescence optics were 469 nm for excitation and 525 nm for emission. All cellular fluorescence images were analyzed using Gen5^TM^ imager software (Ver.3.04, BioTek Instruments, Inc., Winooski, VT, USA).

Fifth, a luminescence was quantified via Caspase-Glo 3/7 assay (Promega Cat. No: G-8091, Madison, WI, USA) to measure Caspase-3/7 enzyme activity as previously described [43]. Briefly, HeLa cells irradiated with LED for 10 min, following incubation with 20 μg/mL of FCF NPs for 2 h, were further incubated for 1, 2, 4, and 8 h. At the end of each incubation period, cells in each well were washed with DPBS, treated with 200 μL of Caspase-Glo 3/7 reagent, mix well using a shaker at 350 rpm for 30 s, and further incubated for 30 min at 37 °C under 5% CO_2_ and 95% air in the dark. Measurement of luminescence in each sample was performed using a multi-mode microplate reader (Cytation 3, Winooski, VT, USA).

Finally, single-cell gel electrophoresis was conducted to detect DNA damage in single HeLa cells (Comet assay, Trevigen, Inc., Gaithersburg, MD, USA). Briefly, HeLa cells were treated with PDT for 10 min and then incubated for 12 h. Cells were collected with a cell scraper, centrifuged, washed with DPBS, and embedded in 1% low-melting agarose (LMA). Then, the cells were immobilized on 2-well Comet slides. The slides were then immersed in Lysis Buffer for 35 min at 4 °C, followed by 30 min incubation at RT in alkaline unwinding solution. Electrophoresis was performed for 30 min at 4 °C and about 1 volt/cm^2^ using a Bio-Rad electrophoresis system (Bio-Rad, Hercules, CA, USA). Then, the slides were neutralized with distilled water and 70% ethanol. The LMA on the slides were completely dried before staining the DNA with SYBR Gold solution (Molecular Probes, Inc., Eugene, OR, USA). The slides were then imaged by an automated live-cell imager (Lionheart FX) with a 2.5x objective lens and fluorescence optics (excitation/emission at 469/525 nm, respectively). The percent DNA in the tail and tail moment were calculated with Gen5 imager software (Ver. 3.04, Winooski, VT, USA).

### 3.8. Statistical Analysis

Quantitative data (n = 4) are expressed as the mean ± standard deviation (S.D.), and significant mean differences were estimated via Student’s *t*-test. Statistical significance was set at *p* < 0.05.

## 4. Conclusions

In the current study, we verified the effect of simple FCF NP surface modification on their in vitro capabilities as theragnostic agents for PDT. The FCF NPs enabled good visualization via MRI and excellent fluorescence cellular imaging in various cancer cells. Furthermore, they showed excellent anticancer activities, resulting in apoptotic cell death via morphological changes in the cell membrane, nuclear, and DNA damage, as well as via significant overexpression of apoptosis-related genes, such as ZFP36L1, CYR61, GADD45G, caspases-2, -3, -9, 10, and -14. The apoptotic cell deaths in cancer cells were dependent on the duration of LED irradiation and FCF NP dose, regardless of cell type. The effects of image-based PDT were attributed to the properties of FCF NPs, such as superparamagnetism, good fluorescence intensity, and production of high singlet oxygen quantum yields from Ce6, as previously described [39]. Thus, the FCF NPs used in this study are most useful for image-based PDT applications. Further in vivo animal studies may be needed to verify the biological and physiological activities of FCF NPs prior to use in clinical trials.

## Figures and Tables

**Figure 1 cancers-12-00571-f001:**
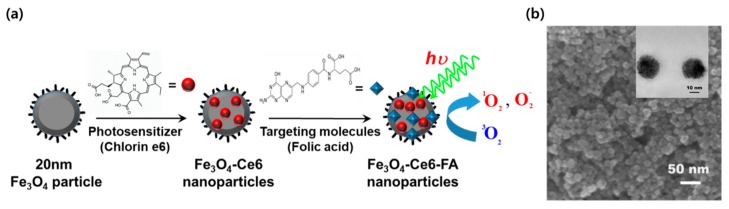
Scheme of synthesis procedure and electron microscope images of multifunctional multifunctional magnetic nanoparticles (FCF NPs). (**a**) Scheme of fabrication procedure, (**b**) images of field-emission scanning electron microscope (FE-SEM) and TEM (small image).

**Figure 2 cancers-12-00571-f002:**
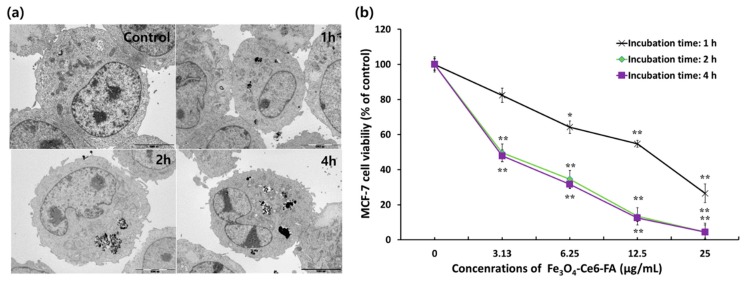
Cellular uptake images and anticancer activities of FCF NPs following photodynamic therapy (PDT) in MCF-7 cells. (**a**) Transmission electron microscope (TEM) images of FCF NPs in MCF-7 cells. TEM images of MCF-7 cells were taken after 1, 2, and 4 h of incubation with 20 μg/mL of FCF NPs. Scale bar = 5 μm. (**b**) Anticancer activities of FCF NPs as a function of incubation time of these nanoparticles (NPs) in MCF-7 cells.

**Figure 3 cancers-12-00571-f003:**
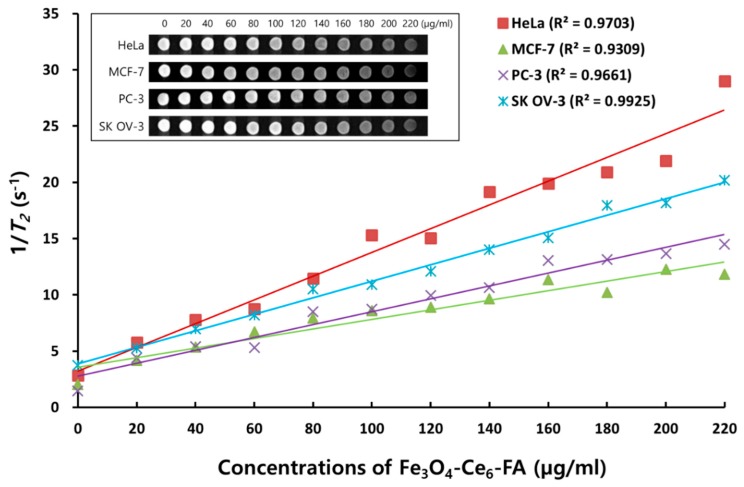
The T_2_ relaxation rate and magnetic resonance (MR) images of FCF NPs in various cancer cells. Each cancer cell was treated with different FCF NP concentrations and incubated for 2 h in the dark. MR images were acquired with a 3.0 Tesla MRI instrument (SIEMENS MAGNETOM Skyra, A Tim + Dot system). The inner box shows the T_2_-weighted MR images of each cancer cell. HeLa: cervical adenocarcinoma; MCF-7: breast adenocarcinoma; PC-3: prostate adenocarcinoma; and SKOV-3: ovary adenocarcinoma cells.

**Figure 4 cancers-12-00571-f004:**
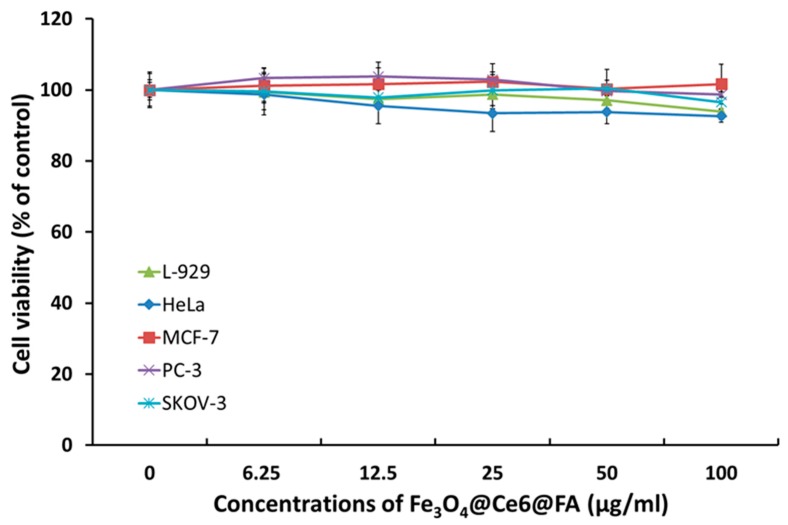
In vitro cytotoxicity of FCF NPs in various cell lines. Cytotoxicity of FCF NPs in L-929 (normal fibroblast), HeLa (cervical adenocarcinoma), MCF-7 (breast adenocarcinoma), PC-3 (prostate adenocarcinoma), and SKOV-3 (ovary adenocarcinoma) cells. All cells were transfected with GFP. The viability of each cell type was detected using a CCK-8 kit following incubation with FCF NPs for 24 h under dark conditions. Data for cytotoxicity are expressed as a mean ± S.D. (n = 4).

**Figure 5 cancers-12-00571-f005:**
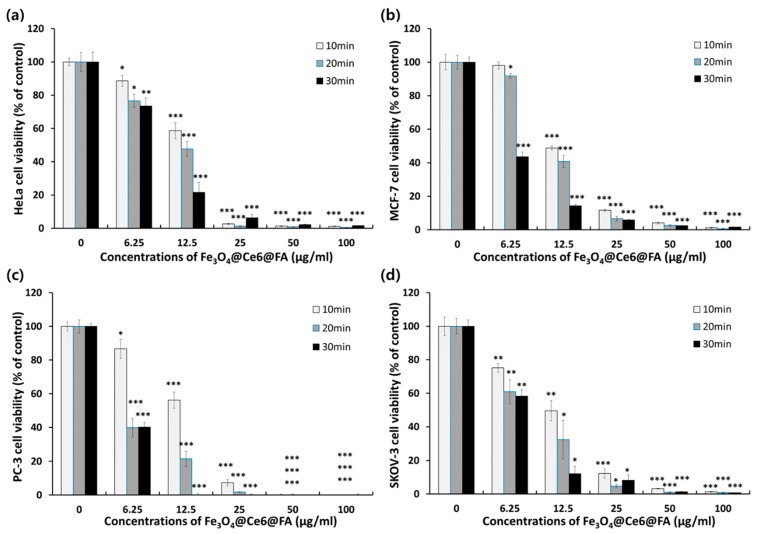
Photodynamic anticancer activities of the FCF NPs in various cancer cells. Phototoxicity of (**a**) HeLa (cervical adenocarcinoma), (**b**) MCF-7 (breast adenocarcinoma), (**c**) PC-3 (prostate adenocarcinoma), and (**d**) SKOV-3 (ovary adenocarcinoma) cells as a function of the duration of exposure to light. Data for photodynamic anticancer activity are expressed as a mean ± S.D. (n = 4), and statistical significance, as analyzed by Student’s *t*-tests, was set at *p* < 0.005 (* *p* < 0.005, ** *p* < 0.0005 vs. control).

**Figure 6 cancers-12-00571-f006:**
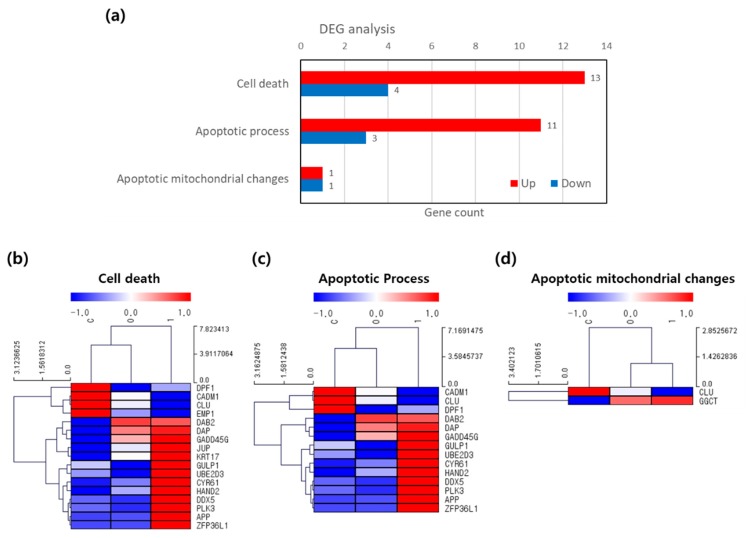
Analysis of mRNA expression related to cell death and apoptotic process in HeLa cells. (**a**) Differentially-expressed gene (DEG) analysis. (**b**) Clustering heatmap for cell death, (**c**) apoptotic process, and (**d**) apoptotic mitochondrial changes in HeLa cells. DEG analysis and mRNA expression were determined by Ampli-Seq sequencing after 40 mW PDT for 10 min and 0 and 1 hr incubation. The fold change (FC) and p-value cutoffs were as follows: FC: > 1.5 and *p*-value: < 0.1.

**Figure 7 cancers-12-00571-f007:**
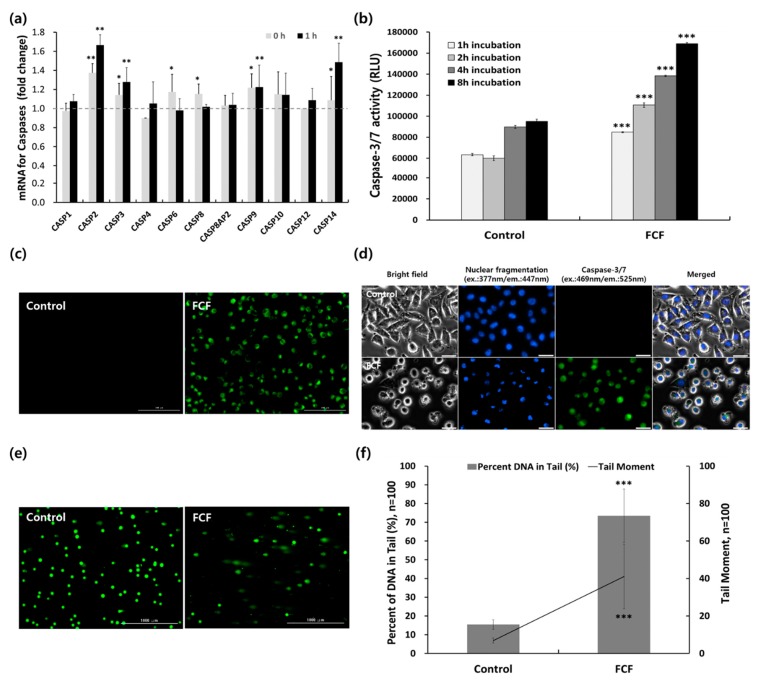
Apoptotic cell death induced by FCF NPs in HeLa cells. (**a**) Caspase mRNA expression was analyzed by Ampli-Seq sequencing after 40 mW PDT for 10 min and incubation for 0 and 1 hr. (**b**) Caspase-3/7 activity over the duration of post-PDT incubation was detected using a Caspase-Glo 3/7 assay kit. HeLa cells were tested 1, 2, 4, and 8 h after 40 mW PDT for 10 min. Data are expressed as mean ± S.D. (n = 4), and statistical significance was set at *p* < 0.05 (*** *p* < 0.0005 vs. control). (**c**) Phosphatidylserine translocation in HeLa cell membranes was stained with fluorescein isothiocyanate (FITC)-conjugated Annexin V (Annexin V-FITC) dye 4 h after 40 mW PDT for 10 min. The green fluorescence signal was produced by Annexin V-FITC. FCF: Fe3O4-Ce6-FA NPs. Scale bar = 100 μm. (**d**) Nuclear fragmentation and Caspase-3/7 activity in HeLa cells. HeLa cells were stained with Hoechst 33342 dye to detect nuclear fragmentation. Caspase-3/7 Green Detection reagent was used to detect Caspase-3/7 activity 4 h after 40 mW PDT for 10 min. Scale bar = 30 μm. (**e**) HeLa cell DNA damage was identified using an alkaline comet assay for detecting DNA damage. (**f**) The percentage of DNA in the tail and tail moment. Quantitative data are expressed as mean ± S.D. (n = 100). Statistical differences were analyzed by Student’s *t*-tests and set at *p* < 0.05 (*** *p* < 0.005 vs. control). Scale bar = 1000 μm.

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
