# Peer review of "Photo-Functionalized Magnetic Nanoparticles as a Nanocarrier of Photodynamic Anticancer Agent for Biomedical Theragnostics"

_cancers, 2020, doi:10.3390/cancers12030571_

Round 1

Reviewer 1 Report

In this manuscript, the authors prepared Fe3O4 nanoparticles functionalized with chlorin e6 and folic acid (FCF NPs). And then they used the FCF NPs for the PDT based theranostic applications. Overall, the experiment of combination of PDT and MRI using this type of magnetic nanoparticle is well designed. However, it has several major and minor issues below. Please address them before publication.

1 The authors have previously reported the magnetic nanoparticle nano-platform. And I understand that more in vitro characterizations were conducted. Yet, I am not quite convinced that compared to the previously published work, enough improvement or novelty were demonstrated in this work. Please elaborate.

2 The writing needs lots of improvements. A) For example, the abstract is not very informative and concise enough to show the impact of the work. B) the introduction did not quite provide a comprehensive overview/background of this research field (e.g. the first paragraph only states some very basic and low-level concept of theranostics). C) The organization of figures: isn’t it better to use figure 7 as the very first figure to give the reader a general idea of the project. Also, some figures need to be enlarged for better visualization.

3 For FA targeting ability, how does the uptake ability of homologous normal cells compare with cancer cells?

4 What are the amounts of Ce6, FA and Fe3O4 in each nanoparticle? How were these calculated?

Author Response

We greatly appreciate the reviewer’s kind and insightful comments regarding our manuscript (Manuscript ID: cancers-723341). According to the reviewer’s comments, the manuscript has been revised. To ease the review process, our responses to specific comments are itemized revised manuscript and changes to the manuscript are highlighted in red.

Reviewer 2 Report

Authors of the paper "Photo-functionalized magnetic nanoparticles as a nanocarrier of photodynamic anticancer agent for biomedical theranostics" presented an interesting research with high novelty. 

Photodynamic methods are still focusing numerous researchers, but the application in clinical trials is unfortunately limited. The approach presented in this study reveals an anticancer potential of photodynamic therapy supported by nanotechnology utilizing magnetic nanoparticles. The proposed nanocarriers are called by authors as biocompatible theranostic agents , what gives an opportunity  in diagnostic imaging. 

In my opinion manuscript can be accepted in the present form.

Author Response

We enthusiastically appreciate the favorable comments expressed about our manuscript (Manuscript ID: cancers-723341) and for considering it worthy of publication in Cancers.

Additionally, we have improved the clarity of the manuscript and the revisions made to the manuscript are shown in red.

Reviewer 3 Report

The manuscript describe an interesting method for applying anticancer reagent. This method will be suitable also for high throughput screening of libraries of compounds.

It was written in a very clear and interesting manner.  

I found a small mistake in line 450-I think it should be Caspase 3/7 and not caspase 2/7.

Author Response

We greatly appreciate the reviewer’s kind and insightful comments regarding our manuscript (Manuscript ID: cancers-723341). According to the reviewer’s comments, the manuscript has been revised. To ease the review process, our responses to specific comments are itemized below and changes to the manuscript are highlighted in red.

Comments:

The manuscript describes an interesting method for applying anticancer reagent. This method will be suitable also for high throughput screening of libraries of compounds.

It was written in a very clear and interesting manner.  

I found a small mistake in line 450-I think it should be Caspase 3/7 and not caspase 2/7.

Response: Thank you for pointing this out, we amended the typo and changed it to “Caspase 3/7”.